# Large-Area Biomolecule Nanopatterns on Diblock Copolymer Surfaces for Cell Adhesion Studies

**DOI:** 10.3390/nano9040579

**Published:** 2019-04-09

**Authors:** Verónica Hortigüela, Enara Larrañaga, Anna Lagunas, Gerardo A. Acosta, Fernando Albericio, Jordi Andilla, Pablo Loza-Alvarez, Elena Martínez

**Affiliations:** 1Institute for Bioengineering of Catalonia (IBEC), The Barcelona Institute of Science and Technology (BIST), 08028 Barcelona, Spain; veronicahortiguela@gmail.com (V.H.); elarranaga@ibecbarcelona.eu (E.L.); alagunas@ibecbarcelona.eu (A.L.); 2Centro de Investigación Biomédica en Red (CIBER), 28029 Madrid, Spain; gerardoacosta@ub.edu (G.A.A.); albericio@ub.edu (F.A.); 3Department of Organic Chemistry, University of Barcelona, 08028 Barcelona, Spain; 4ICFO-Institut de Ciències Fotòniques, The Barcelona Institute of Science and Technology (BIST), Castelldefels, 08860 Barcelona, Spain; Jordi.Andilla@icfo.eu (J.A.); Pablo.Loza@icfo.eu (P.L.-A.); 5Department of Electronics and Biomedical Engineering, University of Barcelona, 08028 Barcelona, Spain

**Keywords:** nanopattern, ligand–receptor interaction, clustering, diblock copolymer, cell adhesion, focal adhesion

## Abstract

Cell membrane receptors bind to extracellular ligands, triggering intracellular signal transduction pathways that result in specific cell function. Some receptors require to be associated forming clusters for effective signaling. Increasing evidences suggest that receptor clustering is subjected to spatially controlled ligand distribution at the nanoscale. Herein we present a method to produce in an easy, straightforward process, nanopatterns of biomolecular ligands to study ligand–receptor processes involving multivalent interactions. We based our platform in self-assembled diblock copolymers composed of poly(styrene) (PS) and poly(methyl methacrylate) (PMMA) that form PMMA nanodomains in a closed-packed hexagonal arrangement. Upon PMMA selective functionalization, biomolecular nanopatterns over large areas are produced. Nanopattern size and spacing can be controlled by the composition of the block-copolymer selected. Nanopatterns of cell adhesive peptides of different size and spacing were produced, and their impact in integrin receptor clustering and the formation of cell focal adhesions was studied. Cells on ligand nanopatterns showed an increased number of focal contacts, which were, in turn, more matured than those found in cells cultured on randomly presenting ligands. These findings suggest that our methodology is a suitable, versatile tool to study and control receptor clustering signaling and downstream cell behavior through a surface-based ligand patterning technique.

## 1. Introduction

In living organisms, cell function is a hierarchical orchestrated phenomenon governed by a multitude of simultaneous cellular processes arising at different spatial and time scales. Cells can interact, recognize, and specifically respond to a vast range of physico-chemical cues [1,2]. Cell membrane serves as the intercommunication system, where cell receptors recognize extracellular ligands, integrate the information and modulate cell response. It is well documented that the spatial arrangement of cell receptors on the cell membrane is a crucial factor controlling the downstream signaling pathways and related functions. Numerous receptors including integrins [3], T-cell receptors [4], N-cadherins, or Eph receptors require assembling into nanoclusters to be functional [5,6]. For instance, in the immune recognition process, a pertinent clustering and reorganization of proteins on the plasma membrane of lymphocytes and antigen presenting cells triggers the formation of the immunological synapse [4]. It is also well established that the spatial arrangement of integrin receptors plays an essential role in the dynamics of the so-called focal adhesion contacts, which rule cell adhesion [7]. Malfunctions in these signaling mechanisms are related to cell functional disorders or even pathological conditions for the organism. Therefore, ruling ligand–receptor molecular interactions will be vitally important for advanced therapeutic and healthcare applications [8,9]. However, the understanding of the underlying mechanisms through which cells interpret the spatial organization of ligand molecules guiding cellular responses remains incomplete [10].

Latest improvements in nanofabrication drove the blooming of numerous nanopatterning techniques with sub-100 nm resolution [11,12]. These provide the means to define and control the spatial organization of signaling molecules, thus, opening a door for modeling spatially resolved cell signaling. Relevant findings can be attributed to these spatially-defined ligand-presenting surfaces, including for instance the identification of a maximum integrin distance leading focal adhesion formation on stiff surfaces [13], the influence of adhesion ligand density on cell migration [14], and the relevance of the local ligand surface density versus the global ligand surface density [15]. In most cases, these studies address the cell response to univalent ligands under different nanometric spatial configurations. However, the signal transduction often depends on multiple receptor–ligand recognition events that can be promoted by a multivalent ligand configuration. Some theoretical models have predicted the multivalent adhesive ligands distribution on the surface, but lack of an experimental characterization of the ligand presentation [16,17,18]. In this context, nanopatterning techniques that allow for a precise spatial of univalent but also multivalent ligands would be instrumental.

Block copolymers (BCPs) are molecules comprised of two or more immiscible polymer fractions joined end to end by a covalent bond. Block copolymers can undergo phase separation due to the immiscibility between covalent-linked blocks, self-assembling in spatially segregated nanodomains of dimensions ranging from 5 to 200 nm [19]. These nanodomains can be easily tuned in size and spacing according to BCP relative composition, the chain molecular weight, the chain architecture, and the fabrication conditions [20]. Compared to other nanopatterning techniques that need sophisticated instrumentation such as electron beam lithography or nanoimprint lithography, block copolymer-based nanopatterning is rapid and inexpensive [21]. Large area samples with a precise spatial disposition can be easily produced, giving an exceptional usefulness to this approach in applications such as high-density information storage devices or solar cells [20]. When coming to biological applications, self-assembled diblock copolymer nanostructures offer exciting opportunities to create surface-bound biomolecule patterns with nanometric resolution. A great variety of geometries can be accessed acting as templates to be replicated by the bioactive molecules [22,23,24].

In here we present a platform based on diblock copolymer surfaces that can be used to produce nanopatterns of ligands over large areas to study multivalent ligand–receptor interactions and their effects on cell-receptor signaling. Diblock copolymers of polystyrene-block-poly(methyl methacrylate) (PS-b-PMMA) that self-assemble into hexagonal arrays of PMMA cylinders embedded in a PS matrix were selected [25]. When deposited as thin films on surfaces that are energetically neutral for both PS and PMMA blocks, PMMA cylinders are oriented perpendicular to the substrate [26,27]. In this way, nanostructures of PMMA circular nanodomains with uniform size and even distribution across the surface were produced. These structures served as templates to anchor ligands and produce biomolecular nanopatterns through the selective functionalization of the PMMA polymeric domains. In a proof-of-concept application, we recently published the success of this approach for a particular block-copolymer composition and ephrin ligands [28]. This time, aiming to prove the versatility of the nanopatterning technique developed, peptides bearing cell-adhesive Arg-Gly-Asp (RGD) moieties were used on two different block copolymers to produce RGD nanopatterns of different dimensions and spacings. In addition, random copolymers of similar composition were used to produce substrates with random ligand disposition at the nanoscale. All substrates had similar global ligand density, but locally this was modified. This way, we were able to screen and discriminate the influence of local ligand density and ligand distribution on the cell behavior. Ligand distribution at the nanoscale was characterized by atomic force microscopy (AFM) and direct stochastic optical reconstruction microscopy (dSTORM) [29]. NIH/3T3 mouse embryonic fibroblasts were cultured on these substrates and cell morphology and focal adhesion formation were characterized. It was observed that cell spreading was rather not affected by the local presentation of ligands when the global surface density was equivalent. Conversely, the spatial distribution of ligands showed a remarkable impact on focal adhesion formation, where the nanopatterned presentation of surface-bound ligands enhanced the maturation of focal adhesions. These findings suggest that ligand presentation in a clustered format promotes multivalent ligand–receptor interactions, therefore altering the cell response. As nanoscale multivalent interactions can potentially activate or modulate cell response, we believe that our biomolecular nanopatterned platform can find applications in systematic studies of ligand–receptor interactions, and in designing new biomaterials or drug-delivery systems.

## 2. Materials and Methods

### 2.1. Fabrication of Nanostructured Block Copolymer Thin Films

Cylinder forming asymmetric diblock copolymers PS-b-PMMA 123-35 (PS molecular weight 123,000 kg/mol, PMMA molecular weight 35,000 kg/mol, polydispersity index 1.09, PS fraction 0.78) and PS-b-PMMA 46-21 (PS molecular weight 46,000 kg/mol, PMMA molecular weight 21,000 kg/mol, polydispersity index 1.09, PS fraction 0.69) were purchased from Polymer Source Inc. (Montreal, QC, Canada) and used without further purification. A random copolymer PS-r-PMMA (molecular weight 14,000 kg/mol, polydispersity index 1.24, PS fraction 0.76) presenting comparable styrene fraction, was also purchased (Polymer Source Inc., Montreal, QC, Canada) to be used as control for non-patterned surfaces. Polymer solutions at different concentrations were freshly prepared for each experiment by dissolving the appropriate amount of polymer powder in 5 mL of anhydrous toluene (Sigma-Aldrich Química, Tres Cantos, Spain). Solutions were stirred for 2 h at room temperature and then filtered (Millex^®^ syringe filter unit, Sigma-Aldrich Química, Spain).

Thin films of diblock copolymers and random copolymer solutions were deposited by spin coating (Laurell Model WS-400A-6TFM/LITE, Laurell Technologies Corporation, North Wales, UK) on top of surface modified glass coverslips (18 mm diameter, Neuvitro, Vancouver, BC, USA). Surface modification was performed by deposition of a thin layer of a random hydroxyl-terminated copolymer brush (PS-r-PMMA-α-hydroxyl-ω-tempo (molecular weight 15,500 kg/mol, polydispersity index 1.15, polystyrene fraction 0.71) also purchased from Polymer Source Inc. (Montreal, QC, Canada). The random copolymer brush was dissolved in toluene at a concentration of 2.5 mg/mL, at room temperature and under stirring for 15 min and spun-coated at 3000 rpm for 40 s. Before thin film deposition, the glass coverslips were cleaned with Piranha solution (1:3, *v*/*v*, H_2_O_2_:H_2_SO_4_) and activated 2 min in O_2_ plasma (Expanded Plasma Cleaner PDC-002 plasma cleaner, Harrick Scientific Corporation, New York, NY, USA). Caution: piranha acid is a strong oxidizer and a strong acid. It should be handled with extreme care, as it reacts violently with most organic materials. The thin films of the random copolymer brush were further annealed in a vacuum oven (VacioTem-T Selecta Oven, Barcelona, Spain) at 220 °C for 7 days. Temperature is a critical parameter in this process, so a Spot Check^®^ surface thermometer (PTC Instruments, Los Angeles, CA, USA) was placed in contact with a representative sample surface to monitor sample temperature during each thermal annealing cycle. Annealing process results in the hydroxyl end-functional groups of the copolymer brush chains diffusing on the glass surface and reacting with the silanol groups [30], resulting in a thin brush layer which shows no energetic preferential affinity for either PMMA or PS blocks. On top of these brush layers, thin films of the two PS-b-PMMA block copolymers and thin films of the PS-r-PMMA random copolymer were deposited, also by spin coating. Subsequently, the block and random copolymer thin films were then thermally annealed at 220 °C for 3 h under vacuum. Once fabricated, the samples were stored at room temperature until further use.

### 2.2. Characterization of Nanostructures on Block Copolymer Thin Films

The surface coverage and thin film thickness of the random copolymer brush layer was characterized by atomic force microscopy (AFM) in a Dimension AFM instrument (Veeco Instruments, Plainview, NY, USA). Measurements were carried out in tapping mode, employing rectangular silicon AFM tips (Nanosensors, PPP-NCHR, spring constant 42 N/m, resonance frequency 330 kHz radius of curvature about 10 nm, aluminum backside coating, and 125 μm in length). The brush layer thickness was determined on on-purpose made scratches. A sharp instrument (razor blade or splinter tip tweezers) was used to perform a scratch in the polymer film, topographic images were obtained, and film thickness was determined from the difference in height found in the edge of the scratch between the mean surface plane and the scratch below. Prior to these measurements, the remaining non-grafted polymer chains of the layer were removed by immersion in 5 mL of fresh toluene for 30 s under agitation. Finally, substrates were blown dried under a pressurized nitrogen flow. The surface morphology, roughness and film thickness of the two block copolymers and the random copolymer were also analyzed using AFM. Thin film thicknesses were again determined from scratch tests. At least three samples and three regions per scratch were imaged to obtain statistical meaningful values. Root mean square (RMS) roughness calculations were performed on 4 µm^2^ images acquired from randomly selected areas over at least three samples from independent experiments. On the two diblock copolymers, the self-assembled periodic structures resulting were characterized by measuring the PMMA cylinder diameter (Ø) and the interdomain spacing (L) (considered as the distance between the nearest cylinder neighbors) on the AFM images. WSxM free software was used to process (simple flatten) and analyze all AFM images [31].

The perpendicular orientation of PMMA cylinders after annealing on the two diblock copolymer thin films was evaluated by AFM and scanning electron microscopy (SEM). For this purpose, the PMMA domains of the block copolymer thin films were selectively etched. First, samples were UV-irradiated for 90 min with a high intensity mercury vapor lamp (UV/Ozone cleaner ProCleaner^TM^, BioForce Nanoscience Inc., Ames, IA, USA). Samples were placed on a Petri dish directly on the stage of the system, at a distance of ~8 mm from the UV source. Then, PMMA degradation products were removed by immersion of the samples in 5 mL glacial acetic acid (17.4 M) (Sigma-Aldrich Química, Spain) for 15 min under agitation. A final rinsing with fresh glacial acetic acid was carried out before samples were blown dried with a pressurized nitrogen flow. Etched thin films were then imaged by a NOVA NanoSEM 230 scanning electron microscope (SEM) (FEI, Eindhoven, The Netherlands). An acceleration voltage of 10 keV in high-vacuum mode was set and secondary electron images were acquired with a Through the Lens Detector (TLD). The nanoporous structures were additionally characterized by topographical images acquired by AFM in tapping mode as previously described in this section. Images were analyzed using ImageJ free software (http://rsb.info.nih.gov/ij, National Institutes of Health, Bethesda, MD, USA). Fast Fourier transformation (FFT) was performed on representative 4 µm^2^ images to illustrate the hexagonal order found on the nanostructures. In addition, defects on the crystal-like structures were evaluated by plotting the Voronoi diagrams corresponding to AFM images. Voronoi plots were obtained by applying the Delaunay/Voronoi plugin of ImageJ software. From the Voronoi diagrams obtained, the density of defects with respect to a perfect crystalline hexagonally packed structure was computed for each block copolymer.

Additionally, surface wettability of the diblock copolymers and the random copolymer surfaces was evaluated. Static water contact angles (WCAs) were measured by the sessile-drop method with an OCA contact angle system (Dataphysics, Filderstadt, Germany). Droplets of 1 µL of water (Milli-Q ultrapure water, Merck Millipore, Madrid, Spain) were dispensed with a syringe. Images of the droplets contacting surfaces were immediately recorded after droplet stabilization (approx. 5 s). The droplet profile was acquired and fitted with SCA20 software (Dataphysics, Germany) applying elliptic fitting method. The resulting values were compared with the ones obtained from pristine PS sheets (Goodfellow, Wrexham, UK) and PMMA surfaces. Flat PMMA films were generated by spin casting a 950PMMA 11% solution in anisole (950PMMA A Resist from MicroChem Corp., Westborough, MA, USA) onto PMMA sheets of 500 µm in thickness (Goodfellow, UK). Mean values of static water contact angles were obtained from measuring at least three droplets per sample on a minimum of three samples per surface category.

### 2.3. Selective Hydrolysis of PMMA

PMMA selective hydrolysis was used to generate carboxylic groups able to react with amine residues of proteins for their covalent binding to the surface. The two block copolymers and the random copolymer were hydrolyzed by immerse them in 10 mL of sodium hydroxide (Sodium hydroxide pellets from Panreac Química S. A. U., Barcelona, Spain) 2 M aqueous solution at 40 °C under cautiously stirring. Different hydrolysis times were tested ranging from 30 min to 5 h to find the optimal conditions preserving thin film integrity [28,32]. After hydrolysis, surface carboxylic groups were protonated with 5 mL of 0.1 M hydrochloric acid solution (Hydrochloric acid 37% from Panreac Química S. A. U., Spain) and rinsed first with MilliQ water and then with absolute ethanol (Panreac Química S. A. U). The integrity and morphology of the hydrolyzed samples was characterized by AFM following the same process detailed for the non-hydrolyzed samples. Attempts to evaluate surface hydrolysis by X-ray Photoelectron Spectroscopy (XPS) and Fourier transform Infrared nanospectroscopy (nano-FTIR) were unsuccessful and lead to inconclusive results. Therefore, the success of the hydrolyzation procedure was established from characterizing the surface distribution of anchored biomolecules.

### 2.4. Biomolecule Functionalization of Nanostructures on Block Copolymers

Biomolecules bearing amine residues were reacted with the carboxylic acid groups selectively generated at the PMMA chains present on the sample surfaces. Prior to biomolecule binding, carboxylic acid moieties were activated by a mixture of N-(3-dimethylaminopropyl)-N-ethyl carbodiimide (EDC) (Sigma-Aldrich Química S. A., Madrid, Spain) (73.4 mg, 0.38 mmol) and N-hydroxysuccinimide (NHS) (Sigma-Aldrich Química S. A., Spain) (8.9 mg, 0.08 mmol) in Milli-Q water (5 mL) at room temperature for 30 min. EDC/NHS chemistry. After reaction, thin films were rinsed with Milli-Q water and absolute ethanol and dried under a stream of nitrogen. Nanostructured templates were then functionalized with amine-containing molecules: Alexa Fluor 647 hydrazide fluorescent dyes (for characterization purposes) and cell adhesive cyclic peptides (RGDfK)-PEG_3_-NH_2_ (for cell adhesion studies).

When functionalized with Alexa Fluor 647 hydrazide fluorescent dyes (Thermo Fisher Scientific, Waltham, MA, USA), samples were incubated with 100 µL of a solution prepared at a concentration of 0.5 mg/mL in Milli-Q overnight at room temperature. After incubation, samples were rinsed with 300 µL of 0.05% Tween^®^ 20 PBS solution and then with PBS three times for 5 min. Then, they were stored at 4 °C until further use. For cell adhesion studies, a cyclic peptide with the integrin-specific amino acid sequence of arginine–glycine–aspartic acid (c(RGDfK)-PEG_3_-NH_2_) was synthesized (yielding purity ≥97% as determined by HPLC analysis). The cyclic peptide structure was coupled with a flexible hydrophilic polymer chain of poly(ethylene glycol)(PEG) end-functionalized with an amine group. The PEG moiety is included as a spacer arm required by surface-bound bioactive molecules to overcome the geometric constrains affecting its functionality. The incorporated primary amine can react with the carboxylic groups exposed by the hydrolyzed surfaces after activation with EDC/NHS. The hydrolyzed diblock copolymers and random copolymer were incubated with 300 µL of c(RGDfK)-PEG_3_-NH_2_ solution at 10 µM concentration in 0.05% Tween^®^ 20 (Sigma-Aldrich Química, Spain) phosphate buffered saline (PBS) for 16 h at room temperature preserving sterile conditions. Samples were then rinsed 3 times with 0.05% Tween^®^ 20 PBS and finally stored at 4 °C in PBS until further use. Non-hydrolyzed thin films were also incubated with 300 µL of 10 µM c(RGDfK)-PEG_3_-NH_2_ solution in PBS for 16 h at room temperature and were used as controls for the selective covalent binding of the amine species to PMMA component.

Samples functionalized with biomolecules were characterized by AFM. Images were acquired in tapping mode in PBS using indistinctly a Dimension AFM instrument (Veeco Instruments, USA) or a Bruker MultiMode 8 AFM (Bruker, Billerica, MA, USA). Microscopes were equipped with triangular silicon nitride AFM probe (Bruker AFM Probes, DNP-S10, spring constant 0.12 N/m, resonance frequency 23 KHz, radius of curvature about 10 nm, reflective gold as back side coating and 205 µm and 40 µm in length and width respectively). WSxM software was used for image processing and simple flatten algorithm was applied previously to height profile acquisition. Flooding algorithm was employer for biomolecule clustering discrimination. At least three topographic AFM images of 4 µm^2^ area were acquired for each functionalization condition assayed.

In addition, samples functionalized with the fluorescent dyes were characterized using a commercial super-resolution NSTORM system from Nikon instruments (Nikon Instruments Europe B.V., Amsterdam, The Netherlands). Laser light at 647 nm was used for excitation through an oil immersion 100x TIRF 1.49NA objective. The emitted fluorescence was then captured by an electron multiplying charge coupled device (EMCCD) camera at an exposure time of 100 ms per frame. Sample imaging was performed in a 5% glucose solution in PBS in the presence of primary thiol (25 µL of 0.1 µg/mL mercaptoethylamine in 1 M hydrochloric acid solution purchased from Sigma-Aldrich Química, Spain) and an enzymatic oxygen-scavenging system (2.5 µL of glucose oxydase (Sigma-Aldrich Química, Spain)), which enhances photostability and photoswitching properties of the dye [33]. Visual Servoing Platform (ViSP) software and the ThunderSTORM plugin of ImageJ free software were used for image processing (http://rsb.info.nih.gov/ij, National Institutes of Health, USA) [34].

### 2.5. Cell Adhesion Assays on the Nanopatterned Substrates

NIH/3T3 mouse embryonic fibroblasts cell line (ATCC^®^ CRL-1658^™^, ATCC, Manassas, VA, USA) from passage 7 to 11 were expanded in cell culture flasks for 2 days at 37 °C and 10% CO_2_ in growth medium (Dulbecco’s Modified Eagle Medium (DMEM, Invitrogen S.A., Barcelona, Spain)) supplemented with 5% Fetal Bovine Serum (FBS), 1% L-glutamine (Invitrogen S.A., Barcelona, Spain), 1% sodium pyruvate (Invitrogen S.A., Barcelona, Spain) and 1% penicillin-streptomycin (Invitrogen S.A., Barcelona, Spain). NIH/3T3 fibroblast were counted with a Neubauer chamber and seeded at a cell density of 4000 cells/cm^2^ in serum-starving medium.

Diblock copolymers and random copolymer substrates were covalently functionalized with c(RGDfK)-PEG_3_-NH_2_ adhesion peptides as explained previously. This process generated nanopatterns of ligands and randomly distributed ligands. As controls, non-hydrolyzed substrates, with physically adsorbed ligands, were also added to the study. To avoid non-specific protein adsorption of non-functionalized areas (PS areas of polymeric thin films), surfaces were passivated by incubation in Bovine Serum Albumin (BSA, Sigma-Aldrich Química, Spain) at 0.5% in PBS for 30 min.

All substrates were preincubated for 10 min with 500 µL of PBS at 37 °C before cell culture. Cells were cultured for 4 h and non-adhered cells were then carefully washed out with PBS. Adhered cells were fixed with 300 µL of 4% paraformaldehyde (Merck, Kenilworth, NJ, USA) in PBS for 20 min at room temperature, rinsed three times with PBS, incubated with 300 µL of 50 mM ammonium chloride (Sigma-Aldrich Química, Spain) in PBS for 20 min and washed again three times with PBS. For immunolabeling of subcellular structures, cells were permeabilized with 0.1% Triton X-100 (Thermo Fisher Scientific, USA) in 1X (10 mM) Tris-Buffered Saline (TBS) (Sigma-Aldrich Química, Spain) for 15 min at room temperature. Then cells were incubated overnight at 4 °C with Phalloidin-Tetramethylrhodamine B isothiocyanate (TRITC-Phalloidin) (Fluka, Buchs, Switzerland) diluted 1:500 and with primary antibody rabbit anti-paxillin [Y113] (Abcam, Cambridge, UK) diluted 1:200 in 3% donkey serum and 0.3% Triton X-100 in 1X TBS. Washing with 0.1% Triton X-100 in 1X TBS was repeated 3 times followed by an incubation with the blocking solution. Blocking was performed with 3% donkey serum (Jackson ImmunoResearch Europe Ltd., Ely, UK) 0.3% Triton X-100 in 1X TBS for 2 h at room temperature, or overnight at 4 °C. Thereafter, cells were incubated for 2 h at room temperature with the secondary antibody goat anti-rabbit IgG Alexa Fluor 488 (Invitrogen S.A., Barcelona, Spain) diluted 1:500 in 3% donkey serum 0.3% Triton X-100 in 1X TBS. Cells were washed three times with 0.1% Triton X-100 in 1X TBS for 15 min. Finally, cells were incubated with Hoechst 33,258 (Invitrogen S.A., Barcelona, Spain) diluted 1:1000 in 3% donkey serum and 0.3% Triton X-100 in 1X TBS for 15 min at room temperature. Samples were washed with 1X TBS for 15 min three times and coverslips were mounted with Fluoromount-G^®^ (Southern Biotechnology Associates, Inc., Birmingham, AL, USA) before storage at 4 °C.

Cell images were acquired with an Eclipse E1000 upright microscope (Nikon, Amsterdam, The Netherlands) equipped with a Charge-Coupled-Device (CCD) camera, an ultraviolet filter, a FITC filter and a G-2A long-pass filter. Fluorescent microscope images were acquired with MetaMorph software (Molecular Devices, Sunnyvale, CA, USA) and were analyzed using ImageJ free software. Samples were imaged with a 10x objective for adhesion analysis. At least, 10 different random locations over each sample were analyzed. Fluorescent nuclei were counted to determine the density of adhered cells and the percentage of cell adhesion was calculated. Differences in the percentage of adhered cells on covalently and non-covalently functionalized substrates were statistically analyzed using a One-way ANOVA test (Origin software). A 40x objective was used for focal contact (paxillin immunostaining) and cell morphology analysis of cells cultured on covalently functionalized thin films. Cell images were taken at 25 random positions over the sample surface (minimum number of 30 cells per sample). As morphology describing parameters, projected area (area of the selected cell), circularity (a circularity value of 1 indicates a perfect circular cell following the equation: circularity = 4π (area/perimeter^2^)), and solidity (solidity = area/convex area, where convex area is the area of the smallest convex polygon comprising the cell area) were evaluated and plotted with the corresponding standard deviation. The number of focal contacts per cell and described size were also analyzed from fluorescence microscopy images.

## 3. Results

### 3.1. Nanostructured Templates for Biomolecular Nanopatterning on Diblock Copolymer Thin Films

For asymmetric cylinder-forming block copolymers with a given composition, the orientation of the minority block will be perpendicular or parallel to the substrate depending on which conformation presents a lower free energy [35]. A controlled interplay between the degree of confinement and the energy of both the substrate and the free interfaces rely on the successful selection of chemical modification of the surface energies, thin film thicknesses and the appropriate thermal annealing conditions. Herein, following the work by Mansky et al. that demonstrated that random copolymers of PS and PMMA could form brushes that modify and control surface wetting [30], hydroxyl-terminated copolymers were used to form brush layers. The removal of unbound copolymer brushes after annealing using ultrasonic bath [27], resulted in deficiently covered surfaces that yield to featureless, non-ordered diblock copolymer films. On the contrary, gently rising with anhydrous toluene lead to densely-grafted copolymer brush layers of thicknesses ~5 nm and RMS roughness values of 0.34 ± 0.02 nm (Appendix A). On these modified surfaces, diblock copolymer thicknesses can be tuned by adjusting either the film deposition parameters (mainly the spinning speed) or the polymer concentration [36]. We found that differences in the thicknesses of the thin films obtained were not relevant when rising the spinning speed from 3000 to 6000 rpm regarding the polymer concentration tested. Therefore, we assumed the asymptotic region of the spinning curves were already achieved at 3000 rpm and set that speed for experimental convenience. Regarding thin film thickness, it is reported that the degree of confinement imposed by the thin film configuration (relationship between polymer interdomain spacing L_0_ and film thickness t) is an important parameter. In here, we tuned film thickness by modifying polymer concentration from 1.5 to 10 mg/mL. A linear dependence of film thickness in this range was observed (Appendix A). Concentrations of 7.5 mg/mL for PS-b-PMMA 123-35 and 5 mg/mL for PS-b-PMMA 46-21 were selected, those leading to film thicknesses of 45 and 38 nm, respectively. For appropriated comparison, a concentration of 5 mg/mL leading to a film thickness of 39 nm was selected for the random copolymer PS-r-PMMA. Finally, after a vacuum annealing process at 220 °C, AFM pictures of the diblock copolymers showed a regular, hexagonal pattern of perpendicular PMMA cylinders that appear slightly higher (1–2 nm) than the surrounding PS matrix (Figure 1). On the contrary, thin films of the random copolymer appeared featureless and homogeneous. Worth mentioning, the brush layer selected lead to the perpendicular arrangement of PMMA cylinders when spun coated at concentrations ranging from 6 to 7.5 mg/mL. However, the hexagonal arrangement of the self-assembled PMMA cylinders was improved at a polymer concentration of 7.5 mg/mL as demonstrated by the closely-packed and highly ordered pattern generated (Appendix A). The order of the nanostructured diblock copolymers was also investigated by imaging the thin films by SEM and AFM after the selective removal of the PMMA block. Upon exposure to ultraviolet light, PS block crosslinks and becomes stiffer while PMMA monomers are degraded and become soluble in glacial acetic acid [37]. On the non-annealed samples, PMMA etched domains resulted a broad range of pore sizes lacking any regularity over the surface due to lack of their perpendicular orientation. On the contrary, annealed samples demonstrated regular nanodomains and fast Fourier transform (FFT) performed on AFM images resulted in a ring-like pattern characteristic of a polycrystalline structures with randomly oriented grains (Appendix A).

Voronoi diagrams were obtained from AFM images of the nanostructures to compute the number of defects on their hexagonal arrangement [38]. PMMA cylinders were considered as central elements surrounded by polygons with several sides corresponding to the number of neighboring cylinders. For perfectly ordered hexagonal structures, Voronoi diagrams will display hexagons. Topographic AFM images and the corresponding Voronoi diagrams of PS-b-PMMA 123-35 and PS-b-PMMA 46-21 thin films are shown in Figure 2. 5-fold and 7-fold-coordinated cylinders associated to defective lattice sites in the grain boundaries are displayed in grey and black, respectively (Figure 2). The relative number of defects known as the ratio between the number of defects out of the total number of cylinders, is 15.6% (PS-b-PMMA 123-35) and 17.9% (PS-b-PMMA 46-21). Similar degrees of imperfection were experimentally found and predicted by three-dimensional cell dynamics simulations [38]. Long-range ordering with non-defective hexagonal lattice has been demonstrated to be improved by controlled solvent annealing [39], solvent-assisted nanoimprint lithography [40], chemical patterning [38] or topographical constrains [41]. A high degree of long-range order is rigorously required for certain nanotechnology applications such as the fabrication of photonic or plasmonic waveguides, while shorter-range order is generally accepted for biomedical applications involving the generation of biomolecular nanopatterns [42].

The PMMA cylinder diameter (Ø_PMMA_) and the interdomain spacing (L) were measured for both PMMA cylinder forming diblock copolymers (Table 1). Both copolymers self-assemble in PMMA nanostructured domains of 29 nm (PS-b-PMMA 123-35) and 21 nm (PS-b-PMMA 46-21), in agreement with reported sizes of similar systems and their L_0_ [37,43,44]. Their interdomain spacings are 64 nm (PS-b-PMMA 123-35) and 37 nm (PS-b-PMMA 46-21). The larger area of PMMA cylinders of the PS-b-PMMA 123-35 is compensated by the higher density of PMMA cylinders of the PS-b-PMMA 46-21 in such a way that both diblock copolymer thin films display a same percentage of PMMA area over the total surface area (~19%). This consideration is not trivial for the further use of these thin films as templates for biomolecule nanopatterning where the PMMA areas will act as anchoring points for covalent binding. The generated nanopatterned thin films will have the same global surface density of PMMA but different nanoscale distribution. On the other hand, random copolymer (PS-r-PMMA) thin films keep an equivalent proportion of PMMA (~24%) without any structure at the nanoscale, so they will be considered to promote a random distribution of biomolecules at their surface.

As surface wettability greatly influences biomolecule-substrate interactions [45], the static water contact angle of the nanostructured diblock copolymers and the random copolymer was evaluated and compared to pristine PMMA and PS films. As shown in Table 2, pure PS and PMMA substrates are rather hydrophobic with relatively high values for water contact angles. For PS-r-PMMA random copolymer thin films, water contact angle increases with respect to PMMA, as it contains a large fraction of PS (0.76). For diblock copolymer thin films, contact angle values have no statistically significant differences with PS. As both nanostructured and random copolymer thin films are similarly hydrophobic, one should not expect wettability differences to dominate biomolecule-substrate interactions on these substrates.

### 3.2. Generation of Biomolecule Nanopatterns on Diblock Copolymer Templates

Previous studies have reported that PMMA can be hydrolyzed under alkaline conditions, leading the formation of carboxylic acid moieties which can react with amine-containing molecules to form an amide bond [28,32]. Following this strategy, the surface of PS-b-PMMA 123-35, PS-b-PMMA 46-21, and PS-r-PMMA thin films were hydrolyzed. After 0.5 h, 1 h, and 5 h of hydrolysis, the morphology of the films was analyzed by AFM (Figure 3). It was found that, as the hydrolysis progresses, the thin film roughness gradually increases from 0.5 nm at the starting time-point, up to 0.8 nm after 5 h. During the initial minutes, the alkaline solution erases the subtle differences in height between island and the PS matrix visible in the original sample. Extended hydrolysis times (t >> 5 h) caused a noticeable erosion of the upper part of the PMMA cylinders, producing a holey structure (data not shown). From these results and, providing that surface integrity is an essential requirement for the subsequent biomolecular pattern, hydrolysis time was limited to 1 h.

Attempts to evaluate the amount of carboxylic acid generated on the PMMA blocks of the surfaces were unfruitful. Therefore, we proceeded with the functionalization of the nanostructured and random copolymers with two biomolecules and we characterized their distribution on the different substrates. PS-b-PMMA 123-35 thin films were selected to optimize the hydrolysis procedure. After hydrolysis, surfaces were activated by EDC/NHS chemistry and finally incubated with Alexa Fluor^®^ 647 hydrazide during 1 h. Using a fluorescent molecule, the spatial distribution of the ligand can be evaluated by two complementary high-resolution techniques: AFM, analyzing the topographic features of the samples and dSTORM, assessing that AFM topography associates to fluorescent molecules. From all the hydrolysis conditions tested (0.5 h, 1 h, and 5 h hydrolysis time, with/without the addition of 0.05% (*v*/*v*) of Tween 20), the condition involving 1 h hydrolysis time adding the surfactant was selected because it showed maximum surface coverage of the ligand (Figure 4). A dedicated section describing all the conditions tested has been published elsewhere [28], while the data presented here only intends to illustrate how the surfaces of the selected protocol look like in their final status. The addition of Tween 20 has been reported to assist in preventing large molecule aggregates and reducing the non-specific binding on surfaces. However, to accurately discriminate between specific and non-specific binding of the fluorescent molecules onto the PMMA islands, a dedicated set of new experiments should be performed. After 1 h of hydrolysis, a narrow distribution of the signal from the fluorescent dye forming clusters with diameters centered at 28.1 nm was observed. In addition, the abundance of large ligand species (>64 nm, which is the interdomain spacing for this polymer) was significantly reduced by the surfactant addition to less than the 5% of the total number of ligand clusters [28]. The selected functionalization parameters achieved a 97% of small clusters (ligand cluster diameter <64 nm), which accounted for the 90% of the total ligand coverage. In average, 105 ± 19 domains per square micron were decorated with ligand clusters, which means a 37% of functionalization success ratio (with respect to the total number of PMMA islands).

On the other hand, for cell adhesion studies, cyclic(RGDfK)-PEG_3_-NH_2_ peptide ligands were immobilized on PS-b-PMMA 123-35, PS-b-PMMA 46-21, and PS-r-PMMA thin films by the optimized functionalization procedure for covalent-binding previously described. Non-hydrolyzed samples were also incubated with the peptides to check account for physiosorbed species. Samples were imaged by AFM to unveil the ligand distribution over the three polymeric substrates. Figure 5 shows topographic AFM images performed in liquid of typical regions and a representative height profile for each one (denoted by a dashed black line in the image). On the non-hydrolyzed samples, physically adsorbed RGD-ligands barely remained attached to the nanostructured thin films and show no visible clustering (Figure 5A,B). On the hydrolyzed nanostructured thin films, RGD-ligands adopted a dot-like pattern on the PMMA circular areas surrounded by the PS matrix (Figure 5D,E). Despite not all the PMMA domains are successfully functionalized (estimated functionalization areas ranged from 11% to 14% of the total sample area), for those that are, the ligand cluster size is uniform and consistent with the dimensions of the nanopatterned template beneath. PS-b-PMMA 123-35 polymer, which provides a PMMA cylinder diameter larger than PS-b-PMMA 46-21, presents a larger ligand cluster size as depicted in the height profiles. Moreover, the pitch of the height profile after RGD-covalent binding matches that of the original diblock copolymer nanotemplates. Opposite to these nanostructured conformation, PS-r-PMMA thin films generate a non-patterned, random ligand disposition (Figure 5F).

### 3.3. Cell Adhesion Studies on Ligand Nanopatterns

The influence of the functionalization strategy (physical absorption or covalent bond) on cell adhesion was examined on the RGD-tailored substrates. Both functionalization strategies were applied to PS-b-PMMA 123-35, PS-b-PMMA 46-21 and PS-r-PMMA thin films and NIH/3T3 mouse embryonic fibroblasts were cultured on them for 4 h. This time-point was selected as a good compromise to diminish potential differences in initial cell adhesion rates attributed to surface physicochemical properties, before cell-ligand interaction begins (few minutes), and avoid the contribution of cell proliferation and cell matrix deposition [46]. After this time, cell nuclei were stained and counted. The percentages of adhesion on the RGD-covalently bound surfaces range from 70% and 78% (Figure 6). On the other hand, cell adhesion surfaces with physically adsorbed RGDs, was found to be 25%, 36%, and 56% for PS-b-PMMA 123-35, PS-b-PMMA 46-21 and PS-r-PMMA, respectively. These values are significantly smaller than those obtained for the covalently-modified thin films and indicate that the covalent functionalization strategy promotes initial adhesion better than physically adsorbed ligands. Even for PS-r-PMMA random surfaces, where similar ligand distributions were obtained according to AFM measurements (Figure 5C,F), cell adhesion was 15% higher when RGD ligands were anchored to the substrate through an amide bond than just physiosorbed.

On the other hand, the percentages of adhered fibroblasts showed no statistically significant differences when cultured on the substrates covalently modified with RGD ligands. Previous reports described a decrease of cell density when single ligand interspacing was larger than 58 nm [13]. Despite our RGD-functionalized PS-b-PMMA 123-35 substrates presented an interdomain spacing of 64 nm, a negative correlation between ligand spacing and cell adhesion was inefficient. The reason behind this reverted tendency may be the presentation of ligand in nanoclusters, which acted as multivalent entities enhancing integrin binding probability in comparison with single ligand arrays. To gain more insight into cell adhesion and spreading processes on surfaces presenting spatially-constrained ligands, NIH/3T3 fibroblasts were immunostained for the observation of actin fibers and focal adhesion contacts (Figure 7A–C). Cell morphology descriptive parameters (spreading area, circularity, roundness and solidity) were evaluated and results are shown in Figure 7D–F. The mean value of projected cell areas showed no significant differences among the samples. Noticeable, cell spreading was more homogeneous on the PS-b-PMMA 46-21 samples, which showed a reduced variability in values of the projected cell areas. Solidity and circularity of adhered fibroblasts showed no statistically significant differences among the different ligand presenting surfaces. However, an increasing tendency in circularity was observed for the block copolymer surfaces.

Unlike cell morphology parameters, focal adhesion formation exhibited clear differences depending on the spatial distribution of RGD ligands at the nanometer scale. The number of focal adhesion contacts per cell was almost doubled in PS-b-PMMA 46-21 and PS-r-PMMA surfaces when compared with PS-b-PMMA 123-35 surfaces (Figure 7H,I)). Moreover, among surfaces PS-b-PMMA 46-21 and PS-r-PMMA, focal adhesions were significantly more mature (focal adhesion area > 1 µm^2^) on the nanopatterned surfaces. Hence, in agreement with literature [47], focal adhesion formation and maturation is highly influenced by ligand spacing, and the closest ligand spacing presented by PS-b-PMMA 46-21 promoted the formation of focal contacts and their maturation. Roca-Cusachs et al. discovered that integrin clustering caused a 7-fold increase in ligand adhesion strength through two mechanisms: the recruitment of cytoplasmic proteins that stabilized the focal adhesion complex and the increase in lateral integrin interactions [48]. Our experiments show that focal adhesion maturation is favored by an aggregated ligand configuration as demonstrated by the 10% increase in mature focal adhesions found on nanopatterned surfaces compared with a random distribution. These results suggest that multivalent ligand presenting configurations could enhance integrin reattachment, reduce their diffusion and thus, mediate stable integrin clustering at the cell membrane laying the foundation of the focal adhesion complex.

## 4. Discussion

Protein patterning techniques have evolved within the past decades to enable the fabrication of micron- to nano-sized features [9,24,49]. Surface-bound biomolecule patterns have unveiled the role of the ligand spatial organization on cell behavior at the nanometer length scale [50]. In a seminal work, site-directed immobilization of thiol-containing peptides was performed on gold nanoparticle arrays fabricated by diblock copolymer micelle nanolithography. By click reaction, single molecules were covalently bound to nanostructured PS-b-PHEMA thin films and the hexagonal arrangement of the PS fraction was replicated [51]. Also, PS-b-PEO diblock copolymer was chemically modified with a maleimide group to link specifically to the cysteine amino acid of certain proteins [52]. These reported strategies have the common purpose of patterning individual proteins that presumably can bind just one receptor. Conversely, the developed thin films in this work are ideal systems for multiple-ligand patterning in a clustered configuration.

Self-assembled diblock copolymer thin films offer exciting opportunities to create surface-bound biomolecule patterns with nanometric resolution over large areas. A great variety of geometries can be accessed acting as templates to be replicated by their functionalization with bioactive molecules [22,23,24,25]. Domain orientation in thin films of block copolymers depends on energetic state of the free interface and the substrate interface. Preferential interactions of PS or PMMA fractions at any of the interfaces dictate the segregation of the preferred polymer block that will wet that interface. PS block preferentially wets the air surface interface due to its lower surface energy compared with PMMA. On the substrate interface, native silicon oxide layers and glass, with a preferential affinity on polar moieties, promote the formation of a PMMA wetting layer on the surface. Therefore, in our block copolymers, both interface effects eventually dictate the orientation of the PMMA cylindrical domains parallel to the substrate. To obtain PMMA perpendicular structures, this tendency should be reverted applying by the proper engineering of both interfacial energies. Traditionally, the free surface interface has been controlled by solvent or thermal annealing [22,53]. For PS-b-PMMA block copolymers, non-preferential wetting conditions are ensured at high temperatures (ranging from 170 °C to 230 °C) when the estimated surface energies for PS (γPS~29.9 mN/m) and PMMA (γPMMA~30.02 mN/m) are comparable [27,30]. Regarding the substrate interface, self-assembled monolayers [54], random copolymer brushes [30], and more recently graphene patterns have been used [55]. In here random copolymer brushes were used to effectively tailor the surface energetics of the substrates, following a common strategy described in the field [30,35,43]. Hydroxyl-terminated random copolymer brushes were densely packed on the surfaces yielding a ~5 nm thick layer, in agreement with the minimum effective thickness reported to prevent penetration of the diblock copolymer chains into the underlying surface [56,57]. The annealing step performed at 220 °C, which exceeds the glass transition temperature T_g_ of both polymer blocks (T_g_ of PS: 103 °C; T_g_ of PMMA: 115 °C) [5,58], facilitates the diffusion of the brush polymer chains towards the interface and the formation of an end-grafted brush layer. On these surfaces, two asymmetric PS-b-PMMA diblock copolymers were shown to form a dense array of perpendicularly oriented PMMA cylinders of ~21 and ~29 nm in diameter, regularly spaced by distances of ~37 and ~64 nm. As the equilibrium state in a diblock copolymer films strongly depends on their degree of confinement [25], film thicknesses were experimentally adjusted by varying the polymer solution concentrations to obtain well-ordered nanostructures. The perpendicular orientation and order of PMMA cylinders was demonstrated by the selective etching of PMMA domains. Some ambiguous defects in the hexagonal lattice were found which prevent long range ordering by our fabrication strategy. This can be improved by adding additional processing steps such as epitaxy [38], and graphoepitaxy [41]. However, biomedical applications including protein nanopatterning usually tolerate low defect density in the hexagonal lattice [42].

Two strategies can be used for the functionalization and biomolecule nanopatterning formation in block copolymer thin films: incorporation of a functional group into the block copolymer backbone and introduction of the functional group after the thin film deposition and self-assembly. Following the first strategy, we reported in a previous work PMMA-b-PS diblock copolymers that incorporated biotin in the PS block, so biotinylated molecules could be coupled after streptavidin coupling [59]. Although the copolymer was successfully fabricated, the product stability was not optimized for long term applications. In another example, PEO block incorporating a maleimide group were used in polystyrene-block-poly(ethylene oxide) (PS-b-PEO) diblock copolymers to nanopattern small peptides through the maleimide-cystein linkage [52]. Attempts to extend the versatility of this approach have also been reported, as the binding of poly-histidine tagged proteins on polystyrene-block-poly(2hydroxyethyl methacrylate) (PS-b-PHEMA) diblock copolymers mixed with iminodiacetic-terminated PS [60]. Although the opportunities to tailor well-studied block copolymer systems exponentially increases by customizing their synthesis, the consequences on the self-assembled nanostructure are not negligible [12,59]. It often remains difficult to predict the effect in phase behavior and microdomain orientation of small structural modifications of block copolymers. Therefore, post-functionalization strategies addressing the selective immobilization of biomolecules on one of the polymeric blocks have been proposed to be advantageous for preserving the ordered nanodomains. The stability of the protein nanopatterning directly depends on the linking chemistry used to immobilize the molecules on the surface [61]. In here, the selective hydrolysis of PMMA domains under alkaline conditions was used to generate carboxylic groups which, after activation, bind covalently to amine-bearing biomolecules, which encompasses any peptide and protein. Hydrolysis conditions leading to fair biomolecule attachment while preserving the thin film nanostructure and stability was established in a previous study [28]. In this study, we demonstrated that RGD motives can be successfully anchored to the nanostructured surfaces and random copolymers. On the nanopatterned surfaces, AFM measurements of the cyclic(RGDfK)-PEG_3_-NH_2_ peptides showed features of sizes compatible with that corresponding of the PMMA islands. Because of the small dimensions of the peptides (~2 nm), we infer that clusters of peptide ligands allocating a maximum amount of 86–154 units (depending on the dimensions of the PMMA domains) were patterned on the surface. On the other hand, no ligand aggregations were seen on the random copolymer surfaces. However, we observed that these last surfaces supported cell adhesion. As this was not happening onto non-hydrolyzed surfaces, we conclude that random copolymers were also functionalized with the peptides.

Nanopatterning techniques have been reported as an excellent tool to investigate ligand–receptor interactions and downstream signaling with single-molecule resolution [61]. Nanoarrays of individual molecules such as RGD peptides, have shed light into the complex signaling interactions between cells and their microenvironment [13,14,62]. However, numerous receptors require assembling into nanoclusters to be functional [3,4,5,6]. This strategy is found and evolutionarily preserved in nature. For instance, some adenovirus, gain access into cells by presenting clusters of cell-adhesive peptides on their surface that bind to multiple integrin receptors simultaneously, in what is called a multivalent ligand–receptor interaction [63]. Soluble multivalent entities as well as surface-bound clusters of ligands have been used to investigate the complex interactions of ligand–receptor binding events [7,64,65]. In particular, focal adhesion formation has been investigated as a function of the density of adhesive ligands on a surface at the nanometer scale. It has been found that a minimum of six RGD ligands/µm^2^, which corresponds to a theoretical interligand spacing of 440 nm was sufficient to promote cell spreading [14]. In our work, we studied the impact of the presentation of multivalent clusters composed of several tenths of cell-adhesive RGD ligands on the formation of focal adhesions through nanopatterned surfaces based on diblock copolymers. Two nanopatterns and one substrate producing a random ligand distribution were selected in such a way that the total area available for biomolecule binding was equivalent (~25%), therefore the global ligand density on the surface is similar, while at the local level differs. The cell adhesion assays showed that the percentage of adhered cells and the cell projected area was not significantly different between the three RGD-tethering surfaces. These results are in good agreement with the well-accepted idea that cell spreading efficiency is critically determined by the overall surface density of ligands [7,66]. Cell density has reported to be reduced when interligand spacing is larger than 58 nm [13]. However, we did not observe this trend on cells cultured on PS-b-PMMA 123-35 thin films, where the pattern spacing is 64 nm. This different behavior might be an attributed to multivalent ligand–receptor interactions promoted by the clustered presentation of the RGD species. On the other hand, focal adhesion formation varied according to the local surface ligand density, as the number of focal adhesions per cell was almost doubled in the samples with the smaller interligand spacing (PS-b-PMMA 46-21 (37 nm) and random copolymer) in comparison with PS-b-PMMA 123-35 (64 nm). Significant differences in focal adhesion maturation were also found between nanopatterned and non-patterned ligand presenting platforms. Focal adhesions were significantly more mature (FA > 1 µm^2^) on the nanopatterned surfaces, as demonstrated by paxillin staining and were spread onto a surface allocating more than a few hundreds of PMMA domains. These results suggest that our developed strategy for multivalent presentation of ligands promoted the clustering of integrins in an effective way by accelerating the dynamics of the receptor oligomerization process, and therefore the recruitment of protein adaptors required for the formation of mature focal adhesions [67,68,69]. As such, our platform can be a valuable system to understand and control receptor signaling and cell behavior through a surface-based ligand patterning technique.

## 5. Conclusions

Biomolecule nanopatterns over large areas were produced by a simple functionalization strategy of diblock copolymer thin films. Through the selective hydrolysis of PMMA domains, amine-bearing molecules were covalently attached to the surfaces, leading to discrete clusters featuring 20–30 nm in size depending on the diblock copolymer selected. The developed ligand presenting surfaces provide a highly precise and well-characterized tool to study cell receptor processes. The nanopatterned ligand-presenting platforms were used to investigate the role of the spatial distribution of adhesive peptides in cell adhesion and focal adhesion formation. It was observed that cell spreading was rather not affected by the local presentation of ligands when the global surface density was equivalent. Conversely, the spatial distribution of ligands showed a remarkable impact on focal adhesion formation, where the nanopatterned presentation of surface-bound ligands enhanced the maturation of focal adhesions. These findings suggest that ligand presentation in the nanoclustered configuration driven by the process presented here might promote multivalent ligand–receptor interactions that impact cell response.

## Figures and Tables

**Figure 1 nanomaterials-09-00579-f001:**
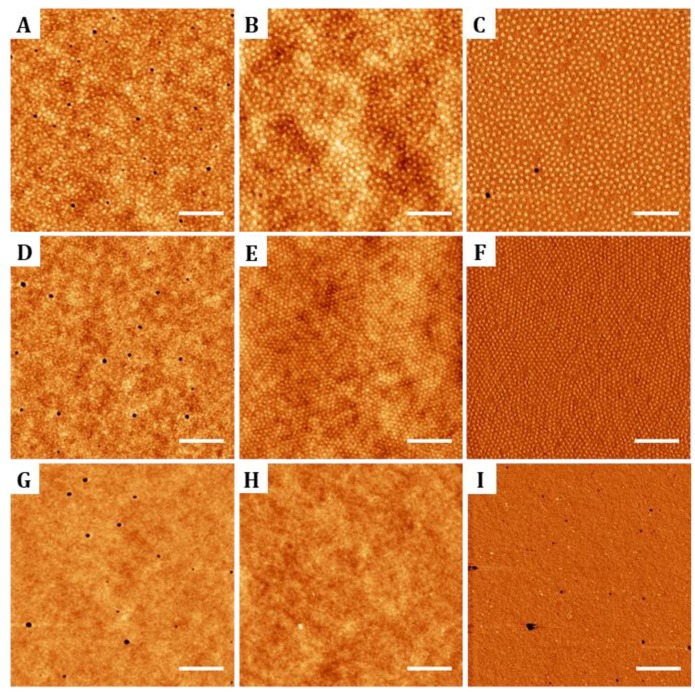
Atomic force microscopy (AFM) images illustrating the topography of thin films of diblock copolymers poly(styrene) (PS)-b-poly(methyl methacrylate) (PMMA) 123-35 (7.5 mg/mL, first row), PS-b-PMMA 46-21 (5 mg/mL, second row) and random copolymer PS-r-PMMA (5 mg/mL, third row). For diblock copolymers, films did not look ordered just after deposition (**A**,**D**) and formed hexagonal patterns after 3 h of thermal annealing at 220 °C: (**B**,**E**) show the sample topography and (**C**,**F**) the corresponding AFM phase images. Panel (**G**) shows the morphology of the random copolymer before annealing, (**H**) corresponds to the topography after annealing and (**I**) is the corresponding phase image. Scale bar 400 nm; Z-scale: 5 nm.

**Figure 2 nanomaterials-09-00579-f002:**
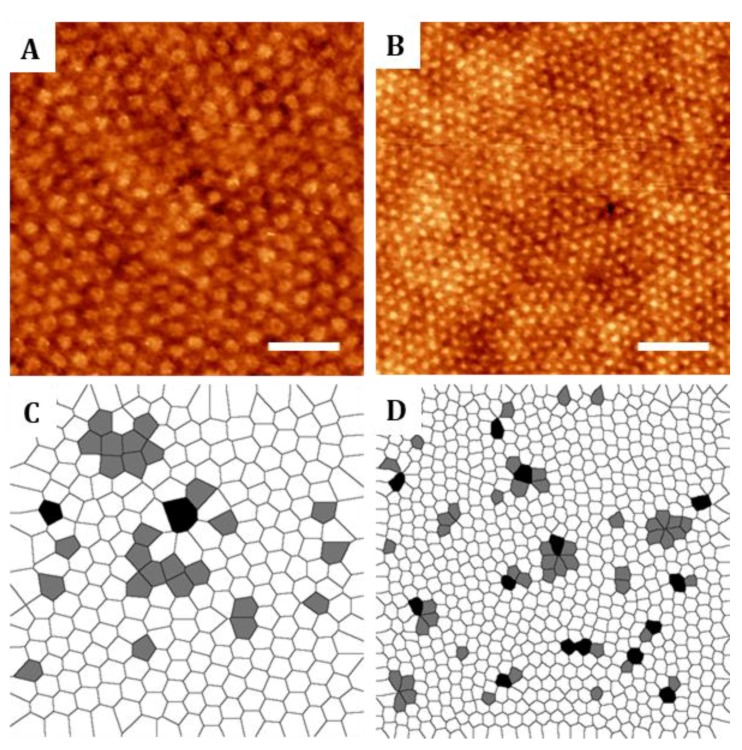
Representative topographic AFM images of ordered thin films of (**A**) PS-b-PMMA 123-35 and (**B**) PS-b-PMMA 46-21 diblock copolymers and (**C**,**D**) corresponding Voronoi diagrams indicating the defects in the ordered structures. Scale bars: 400 nm.

**Figure 3 nanomaterials-09-00579-f003:**
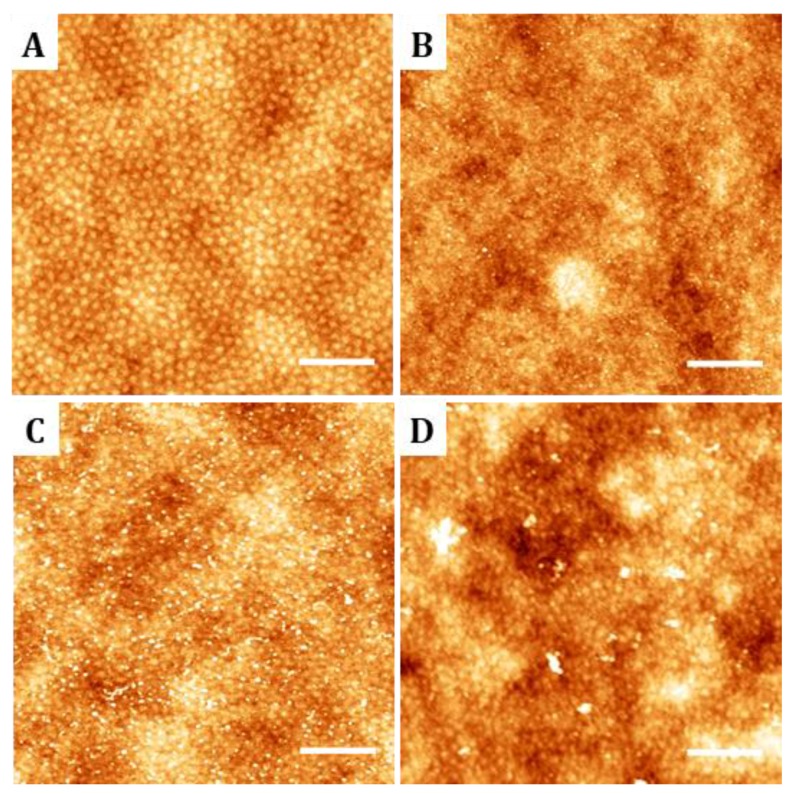
AFM images of the topography of thin films of PS-b-PMMA 123-35 diblock copolymer after hydrolysis reaction in sodium hydroxide 2 M at 40 °C during (**A**) 0 h, (**B**) 0.5 h, (**C**) 1 h, and (**D**) 5 h. Root mean square (RMS) roughness increases from 0.5 nm (0 h) to 0. 8 nm (5 h). Scale bar 400 nm; Z-scale: 5 nm.

**Figure 4 nanomaterials-09-00579-f004:**
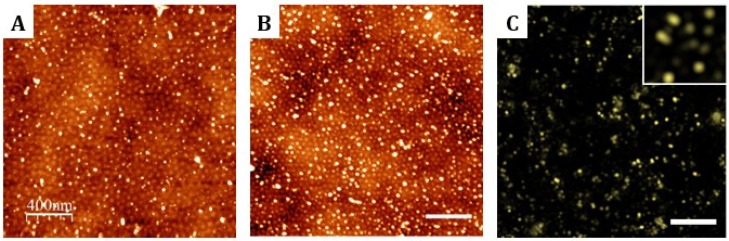
(**A**) AFM images of the topography of thin films of PS-b-PMMA 123-35 diblock copolymer after 0.5 h of hydrolysis and (**B**) after 1 h of hydrolysis and functionalization with Alexa Fluor^®^ 647 hydrazide fluorescent dye. Scale bar 400 nm; Z-scale: 5 nm. (**C**) Direct stochastic optical reconstruction microscopy (dSTORM) image showing the surface distribution of the from the Alexa Fluor^®^ 647 hydrazide dye fluorescence. Inset shows the clusters of the dye following the underneath nanostructured template. Scale bar 400 nm.

**Figure 5 nanomaterials-09-00579-f005:**
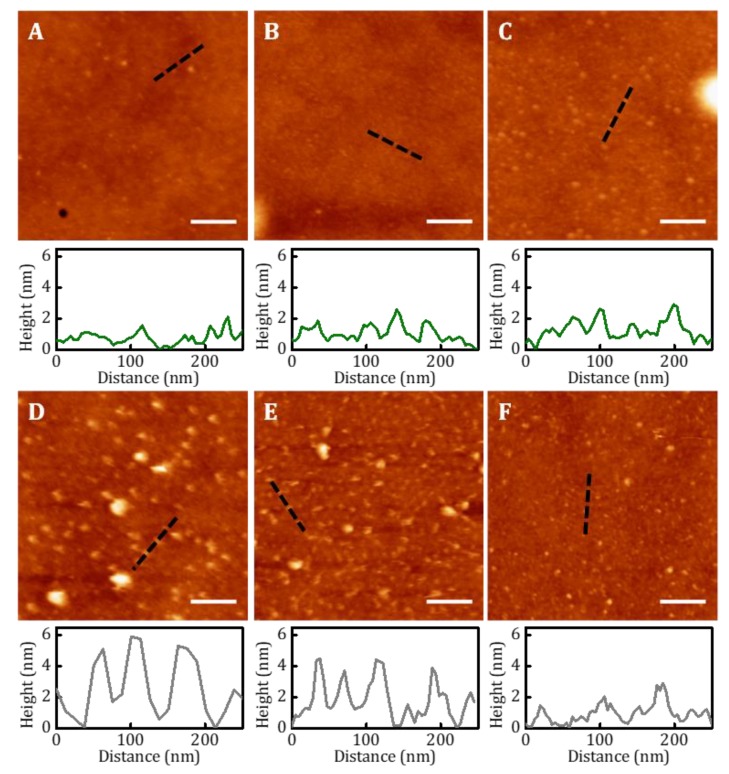
AFM topography images and height profiles (along the black dashed lines) of physically adsorbed and covalently-bound cyclic(RGDfK)-PEG_3_-NH_2_ peptides on hydrolyzed diblock copolymer and random copolymer thin films. Images were acquired in liquid conditions. (**A**) Physical adsorption on PS-b-PMMA 123-35, (**B**) on PS-b-PMMA 46-21, and (**C**) on PS-r-PMMA thin films. (**D**) Covalently bound peptides on PS-b-PMMA 123-35, (**E**) on PS-b-PMMA 46-21, and (**F**) on PS-r-PMMA thin films. Scale bar 200 nm; Z-scale: 20 nm.

**Figure 6 nanomaterials-09-00579-f006:**
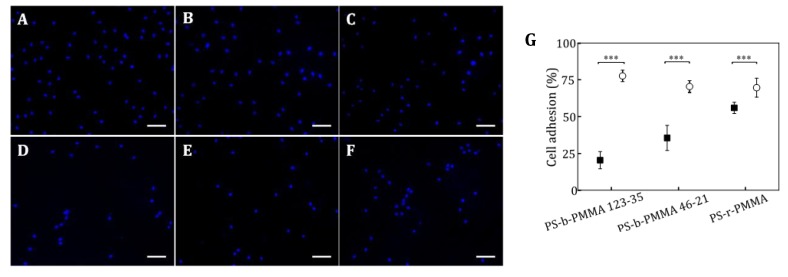
NIH/3T3 mouse embryonic fibroblast cells cultured on cRGDfK-NH_2_-PEG_3_ modified surfaces for 4 h in starving medium. Arg-Gly-Asp (RGD) peptides were either covalently bound or physically adsorbed on diblock copolymer and random copolymer surfaces. (**A**) Fluorescent microscopy image of cell nuclei on a surface with covalently bound peptides on PS-b-PMMA 123-35 surfaces, (**B**) PS-b-PMMA 46-21, and (**C**) PS-r-PMMA. (**D**) Fluorescence microscopy image of cell nuclei on a surface with physical adsorbed peptides on PS-b-PMMA 123-35, (**E**) PS-b-PMMA 46-21, and (**F**) PS-r-PMMA. Staining: Hoechst for nuclei (blue); Scale bar 100 µm. (**G**) Percentage of cell adhered (with respect to the number of seeded cells) on covalently attached (hollow circles) or physically adsorbed (bold squares) RGD-ligands on top of the three different substrates. (*** denotes statistical significance level *p* < 0.001; error bars represent the standard error of the mean).

**Figure 7 nanomaterials-09-00579-f007:**
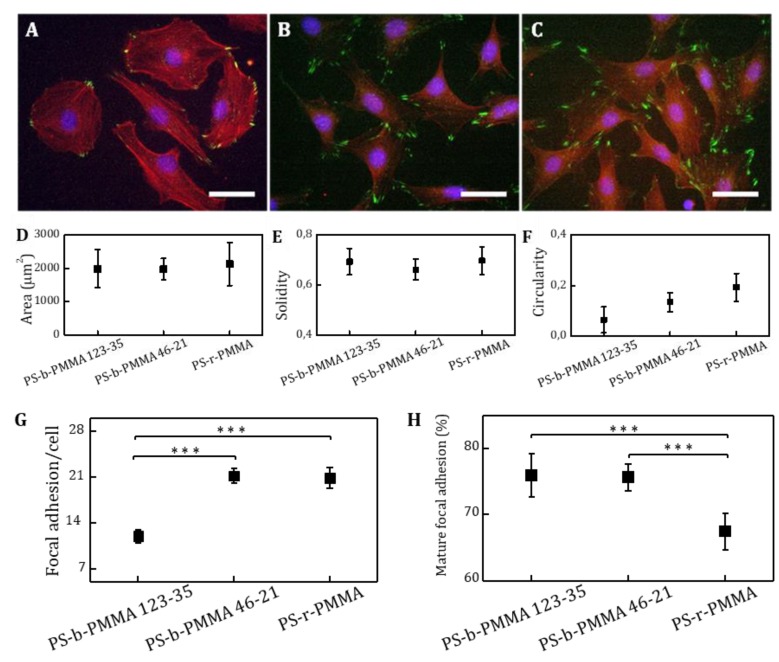
Representative fluorescence microscopy images of NIH/3T3 fibroblast cells plated on (**A**) PS-b-PMMA 123-35, (**B**) PS-b-PMMA 46-21, and (**C**) PS-r-PMMA after c(RGDfK)-PEG_3_-NH_2_ ligand functionalization and BSA passivation under starving conditions (nuclei in blue, actin fibers in red and focal adhesion contacts in green). Scale bar, 25 µm. Morphological parameters of the adhered cells: (**D**) projected cell area, (**E**) solidity, and (**F**) circularity after 4 h of cell. Focal adhesion analysis: (**G**) number of focal adhesions per cell and (**H**) percentage of mature focal adhesions (*** denotes statistical significance level *p* < 0.001; error bars represent the standard error of the mean).

**Table 1 nanomaterials-09-00579-t001:** Characteristic parameters of copolymer thin films. PS-b-PMMA 123-35, PS-b-PMMA 46-21 and PS-r-PMMA were spun coated at 7.5 mg/mL, 5 mg/mL, and 5 mg/mL, respectively. After annealing at 220 °C for 3 h, film thickness (t), RMS roughness (RMS), diameter of PMMA cylinders (Ø_PMMA_), interdomain spacing (L), and density of nanodomains (d) were calculated from AFM images. Values are given as the mean ± standard deviation and are obtained from images of 2 µm x 2 µm in size. At least 3 images per sample from 3 independent experiments of the different polymer compositions were evaluated.

Parameter	PS-b-PMMA 123-35	PS-b-PMMA 46-21	PS-r-PMMA
t (nm)	44.9 ± 0.7	38.5 ± 1.1	39.5 ± 0.9
RMS (nm)	0.8 ± 0.2	0.6 ± 0.1	0.4 ± 0.1
Ø_PMMA_ (nm)	28.5 ± 1.8	21.4 ± 1.6	-
L (nm)	64.4 ± 4.7	36.9 ± 1.1	-
d (µm^−2^)	287 ± 12	526 ± 43	-

**Table 2 nanomaterials-09-00579-t002:** Water contact angle measurements for different substrates. Values are reported as the mean ± standard deviation from at least 5 independent measurements.

Substrate	Water Contact Angle (°)
PMMA	83.2 ± 5.4
PS	96.6 ± 2.6
PS-r-PMMA	89.6 ± 2.0
PS-b-PMMA 123-35	96.8 ± 2.5
PS-b-PMMA 46-21	86.8 ± 3.9

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
