# Peer review of "Large-Area Biomolecule Nanopatterns on Diblock Copolymer Surfaces for Cell Adhesion Studies"

_nanomaterials, 2019, doi:10.3390/nano9040579_

Round 1
Reviewer 1 Report
This paper by Hortiguela et al. investigates the functionalization of nanostructured block copolymer thin film surfaces to systematically modify cell adhesion. Using PS-PMMA block copolymer and modifying the PMMA to attach cell adhesion peptides, it was shown that the focal adhesion maturity is increased on the block copolymer compared to the random polymer. Also, the spacing and the areal density are both important to the cell adhesion results. Overall this is an interesting paper. PS-PMMA thin films are a very well-studied area, but the experiments shown here are in agreement with those previous results. The surface functionalization is more qualitative as direct characterization of the surface functionalization was not possible, but its result is shown in the cell adhesion studies.
My only major comment for revision is the results in Figure 6. It does not appear to agree with the discussion in the text. The images in A,B,C and D,E, F appear that the absorbed surfaces show higher density of cell adhesion. Similarly, the data described by the symbols in Figure 6G appear to be opposite what is written in the text. This needs to be carefully checked.
Minor
The distance from the UV lamp in the UV/ozone cleaner should be given. If it is too close I would think it would etch the PS block
In the PMMA soponification, how is HCl used to neutralize the carboxcyclic acid? Isn't it already in the sodium form from the reaction with NaOH?
Author Response
We thank the reviewer for the positive appreciation of our work and, especially, for the detail in the reviewing process. Below is our point-by-point response to the concerns /questions arised:
Point 1. My only major comment for revision is the results in Figure 6. It does not appear to agree with the discussion in the text. The images in A,B,C and D,E, F appear that the absorbed surfaces show higher density of cell adhesion. Similarly, the data described by the symbols in Figure 6G appear to be opposite what is written in the text. This needs to be carefully checked.
Response to point 1. We really thank the reviewer for realizing about this mistake. As we were working with several versions of figure 6, definitely the figure capture was messed up. We apologize for this mistake, which has been corrected in this new version of the manuscript (page 14).
Point 2. The distance from the UV lamp in the UV/ozone cleaner should be given. If it is too close I would think it would etch the PS block.
Response to point 2. A sentence specifying where exactly the samples were placed in the system has been added to the Material and Methods section (page 4) to clarify this issue.
Point 3. In the PMMA soponification, how is HCl used to neutralize the carboxcyclic acid? Isn't it already in the sodium form from the reaction with NaOH?
Response to point 3. We used “to neutralize” in terms of ionic charges, meaning sodium carboxylate will react with protons from HCl neutralizing its negative charge. We agree with the Reviewer in that this is inaccurate and can be confusing, therefore we have replaced the word “neutralized” by “protonated” in the text (page 5).
Reviewer 2 Report
In this work, the authors demonstrate and characterize a method to fabricate large surface areas with nanopatterned biomolecules to study effects of receptor/ligand interactions. This field has seen many approaches to achieve the spatial patterning of RGD molecules over the last 15 years, and the authors make an effort to describe the pros and cons of their technique in the context of the many alternatives. As I see it, the primary advantage is derived from the self-assembly of patterned features during annealing, leading to reproducible and well-ordered patterns over large areas. The work is clear, well-explained, and includes a thorough physical characterization of the materials. However, a number of remaining ambiguities related to the biological characterizations should be addressed before publication.
Issues to be resolved:
1) The authors should explain why they chose the 4-hour timepoint for the cell analysis. Four hours after seeding, it is difficult to deconvolute the degree of ligand-mediated attachment from the rate of cell attachment, especially given the differences in contact angle, ligand distribution, etc. It could be that the differences in cell morphology are an artifact of different attachment rates. The contact angle of PS‐b‐PMMA 46‐21 is not reported.
2) In Figure 4, it appears that only a subset of the PMMA features are labeled with AF647. The authors should comment on the efficiency/degree of ligand functionalization. Figure 5 also suggests that the RGD ligands are bound to only a fraction of the PMMA features. Figure 4 should also include a non-hydrolyzed surface to further demonstrate that the ligand attachment is specific. The PS‐b‐PMMA 46‐21 condition is also missing from Figure 4. Since we have only circumstantial evidence that the RGD ligands are not attaching nonspecifically to PS regions (e.g. ligand spacing measurements), the authors should consider co-functionalization of RGD and AF647 together and checking for co-localization of AF647 and focal adhesion signal. Otherwise, the reader can only guess that focal adhesions coincide with patterned RGD.
3) The author’s should discuss/estimate how many ligands they expect to be presented by each PMMA feature, and how many features are expected to be bound per mature focal adhesion and per cell.
4) The label fonts in many of the figures are blurry and Figure 2 needs a scale bar.
Author Response
We thank the reviewer for his/her comments and diligence in the review process. Please find below our point-by-point response to his/her queries.
Point 1. The authors should explain why they chose the 4-hour timepoint for the cell analysis. Four hours after seeding, it is difficult to deconvolute the degree of ligand-mediated attachment from the rate of cell attachment, especially given the differences in contact angle, ligand distribution, etc. It could be that the differences in cell morphology are an artifact of different attachment rates. The contact angle of PS‐b‐PMMA 46‐21 is not reported.
Response to point 1. We selected a timepoint of 4 hours, within the first 24 hours of seeding, as it is normally done when surface-induced adhesion is to be evaluated. On one hand, short time points are needed to avoid the contribution of cell matrix deposition and cell proliferation as much as posible [1]. On the other hand, and as highlighted here by the Reviewer, the time-point should be large enough to diminish the effects of initial cell-deposition time, where there is no contribution from surface ligands, before cell-ligand interaction begins. This usually takes several minutes and is expected to be approximately the same for all the evaluated samples, as no dramatic differences in properties such as wettability, are measured. Therefore, authors believe that the changes in cell morphology and focal adhesion maturation observed at 4 hours are barely influenced by the deposition time and indeed essentially promoted by ligand clustering-mediated attachment, which lead to different attachment rates. Indeed, as it has been reported, ligand-receptor interactions are generally discussed in kinetic terms. Surface-induced ligand clustering has been reported to accelerate the dynamics of receptor oligomerization process [2,3], as seen when comparing, at the same global ligand surface density, nanopatterns to surfaces with ligands evenly distributed. When dealing with adherent cells, delayed adhesion translates in the death of the non-adhered cells. Along this line and, as requested by the reviewer, the text in page 13 has been modified to justify the selected timepoint of 4 hours and the discussion in page 17 has been also modified, including relevant references, to address the kinetic effects of ligand clusters on the dynamics of receptor oligomerization.
As requested by the reviewer, also Table 2 has been modified to include the contact angle of PS-b-PMMA 46-21.
Point 2. In Figure 4, it appears that only a subset of the PMMA features are labeled with AF647. The authors should comment on the efficiency/degree of ligand functionalization. Figure 5 also suggests that the RGD ligands are bound to only a fraction of the PMMA features.
Response to point 2. The reviewer is right, only a fraction of the PMMA islands are hydrolyzed with an efficiency that is high enough to allocate a sufficient amount of ligands that are visible either by AFM or by STORM. We have computed that functionalization degree from the AFM pictures and added page 11. This is similar when adding the RGD ligands. As we understand that maybe in that case the relevant parameter is the percentage of surface covered by the ligand, we added also this information to results reported in page 12.
Point 2 (cont.). Figure 4 should also include a non-hydrolyzed surface to further demonstrate that the ligand attachment is specific
Response to point 2 (cont.). The reviewer suggests including a non-hydrolyzed control in Figure 4. In our understanding, this experiment is more relevant with the RGD ligands and that is why these analyses were performed in a systematic way and reported in Figure 5.
Point 2 (cont.). The PS‐b‐PMMA 46‐21 condition is also missing from Figure 4.
Response to point 2 (cont.). In our reasoning, the purpose of the work was to compare all the three hydrolyzed samples for the same hydrolysis conditions. For this purpose, we selected the PS-b-PMMA 123-35 for the optimization of the functionalization procedure, as the size of the features generated (mainly the spacing between the PMMA islands) was large enough to provide the best sensitivity with the imaging techniques employed. Once the hydrolysis conditions were optimized with a fluorophore molecule, we then performed and characterized the samples of our interests, these being the RGD functionalized surfaces (Figure 5).
Point 2 (cont.). Since we have only circumstantial evidence that the RGD ligands are not attaching nonspecifically to PS regions (e.g. ligand spacing measurements), the authors should consider co-functionalization of RGD and AF647 together and checking for co-localization of AF647 and focal adhesion signal. Otherwise, the reader can only guess that focal adhesions coincide with patterned RGD.
Response to point 2 (cont.). The reviewer suggests performing measurements to check the co-localization between the RGD ligands and the focal adhesions. Although extremely interesting, we understand that the nature of the experiments suggested lays well beyond the scope of these manuscript, as is was not our intention, as it is described in the manuscript, to determine co-localization between the ligands and the focal adhesions. Basically, technical problems arise from the fact that mature focal adhesions are very large in size (> 1 µm2) compared to the nanopatterns (~ 0.002 µm2) and can allocate below them theoretically more than 500 PMMA domains. Because we are staining paxillin, they are also in different z-planes. Therefore, it is technically challenging to find a technique suitable to resolve both entities in a single image and it is also difficult to make the proper correlations, as recent publications have demostrated [4]. Co-localization of ligand-receptor in the case of multivalent interactions can be resolved at early time points (when only small receptor oligomers are involved) and in the case of single ligands.
Point 3. The author’s should discuss/estimate how many ligands they expect to be presented by each PMMA feature, and how many features are expected to be bound per mature focal adhesion and per cell.
Response to point 3. Taking into account the estimated dimensions of the RGD ligands employed (~ 2 nm) [5], a maximum of 86-154 ligands/ PMMA nanodomain could be bound, which greatly exceeds the threshold numbers for the minimum amount of ligands required to promote the formation of focal adhesions (> 6) [5]. As explained above, each mature focal adhesion can allocate a large number (more than 500) of PMMA features. We have included this estimations in the discussion section in page 13.
Point 4. The label fonts in many of the figures are blurry and Figure 2 needs a scale bar.
Response to point 4. Scale bar in Figure 2 has been added. We have increased the quality of the pdf text to enhance the resolution of the figures.
[1] Hersel, U., Dahmen, C. & Kessler, H. RGD modified polymers: biomaterials for stimulated cell adhesion and beyond. Biomaterials 24: 4385-4415 (2003).
[2] Liu, J., Liu, M., Zheng, B., Yao, Z. & Xia, J. Affinity enhancement by ligand clustering effect inspired by peptide dendrimers−shank PDZ proteins interactions. PLoS ONE 11: e0149580 (2016).
[3] Benard, E., Nunès, J. A., Limozin, L. & Sengupta, K. T cells on engineered substrates: The impact of TCR clustering is enhanced by LFA-1 engagement. Front. Immunol. 9: 2085. (2018) doi: 10.3389/fimmu.2018.02085
[4] Oria, R., et al., Force loading explains spatial sensing of ligands by cells. Nature. 552, 219-224 (2017). doi:10.1038/nature24662
[5] Arnold, M.; Hirschfeld-Warneken, V.C.; Lohmüller, T.; Heil, P.; Blümmel, J.; Cavalcanti-Adam, E.A.; López-García, M.; Walther, P.; Kessler, H.; Geiger, B.; Spatz JP. Induction of cell polarization and migration by a gradient of nanoscale variations in adhesive ligand spacing. Nano Lett. 2008, 8(7), 2063–2069. https://doi.org/10.1021/nl801483w
Reviewer 3 Report
The manuscript by Hortigüela et al., tests nanopatterning of large biomolecules on diblock copolymer surface for cell adhesion. They formed nanodomains using PS and PMMA co-polymers and conjugated of bioactive peptides on them. They performed AFM, cell adhesion and spreading assays on them to show how these nanopatterning with adhesive peptides of different size and spacing regulate integrin receptor clustering and the formation of cell focal adhesions. The results showed increased focal points and mature cells. This is an interesting method to study signaling and downstream cell behavior. Overall, a well written manuscript with properly designed expts, and nicely elaborated results and discussion.
Author Response
We thank the reviewer for the positive appreciation of our work and her/his diligence during the review process. Spelling errors have been corrected in the text.
Round 2
Reviewer 1 Report
The authors revisions are acceptable.
Author Response
We thank the reviewer for acceting our modifications to the paper.
Reviewer 2 Report
I appreciate the authors' efforts to improve the manuscript and accept their rationale for selecting the 4-hour timepoint for cell analysis. I also accept that microscopic detection of ligand/receptor co-localization may be beyond the scope of this initial characterization of their fabrication method. However, I do not agree that proper controls are unnecessary in Figure 4 simply because non-hydrolyzed samples are evaluated in Figure 5. Without a dedicated negative control sample, the reader cannot know how much of the AF647 signal is actually associated with hydrolyzed PMMA regions. It is very likely that there is some degree of non-specific adsorption, and the extent of this must be shown.
Author Response
We thank the reviewer for her/his comments and suggestions to improve our manuscript. We also strongly agree with her/his comment on the presence of non-specific adsorption on the samples, which can be minimized but not completely avoided. Unfortunately, we disagree with the control she/he suggested to evaluate this non-specific adsorption.
On one hand, any possible even non-specific adsorption of AF647 in Figure 4 cannot be detected by neither of the techniques employed in this study because of the small size of the AF647. Actually, it is quite unlikely that PMMA domains that do not show a clear signal of the AF647 molecule neither by AFM nor by STORM (lateral resolution around 20 nm), have actually any of the AF647 molecules adhered, either specifically or non-specifically. However, it is only when the molecules are close enough to form clusters that can be resolved in a faithful manner by the employed techniques.
We emphasize that the aim of this figure was to illustrate the time point selected for the hydrolysis of the samples,, as shorter time points lead to samples with fewer PMMA domains covered.
Therefore, since we observed differences in the amount of PMMA domains covered at different hydrolysis times, we consider this as a direct effect from covalent bonding. To better illustrate this point, a sample hydrolyzed half of the time and characterized by AFM has been included in Figure 4. In a previous work published by the group, a dedicated section on the optimization of the hydrolysis process can be found. As maybe this was not obvious from reading the text, we included a sentence calling this reference in the proper section (page 11).
Still, we strongly believe that we cannot extrapolate the results of the non-specific adsorption of the AF647 to the RGD samples, where we believe are the relevant samples to evaluate these effects, as they are the ones used to culture the cells. RGD system is not equivalent to the AF647. On one hand, the molecules are different, so they will non-specifically adsorb to the sample in a different way. On the other hand, it is the adhesion rate of the cells what actually would determine the degree of adsorbed RGD ligands in practice, as we see that cells, even with if they are passivated with BSA after incubation with the RGD ligand, still adhere to the surface to some extent.
Round 3
Reviewer 2 Report
In the current form, Figure 4C has no control to allow the reader to differentiate the signal contribution of specific AF647 binding to hydrolyzed PMMA from false-positive signal derived from non-specific adsorption elsewhere on the surface (e.g. on non-PMMA or non-hydrolyzed regions). The way to demonstrate specific AF647 signal would be to repeat the experiment on a non-hydrolyzed surface. The dSTORM technique can absolutely resolve single AF647 molecules, but this is not required. The negative control simply needs to have the same localization precision and optical resolution as the image provided for the hydrolyzed sample in 4C.
The authors added an AFM image in Figure 4A with a 30 minute hydrolysis. The same could and should be done for the dSTORM fluorescence image. An analogous 30-minute hydrolysis sample would add some insight, and a non-hydrolyzed (0 minutes) control would allow the reader to fully subtract the baseline level of non-specific signal. Analogous controls are provided for the RGD system in Figure 5, but as the authors mention this is a different scenario and cannot be directly compared. I am not asking the authors to extrapolate the AF647 binding result to RGD samples, I am asking for internal controls for the data presented in Figure 4 so that the reader knows that the signal shown correlates to hydrolyzed PMMA that is labeled with dye and not nonspecific adsorption.
Author Response
We thank the reviewer
for his/her comments and suggestions. We understand that, despite he/she
acknowledges that data of Figure 4 can not be directly extrapolated to Figure 5
at this point what he/she is requiring completeness of the data presented.
Unfortunately, both time and personnel resources prevent at this moment of
performing a new set of experiments that would be conforming a new figure with
the complete study required. As this is not possible at this stage, we have
modified the text associated to Figure 4 in page 11, 12 and in the discussion
in page 17 to acknowledge the limitations of the data currently shown and that, as
suggested by the reviewer, further controls will be needed to discriminate
between specific and non-specific adsorption.